# Antimicrobial Peptides and Physical Activity: A Great Hope against COVID 19

**DOI:** 10.3390/microorganisms9071415

**Published:** 2021-06-30

**Authors:** Sonia Laneri, Mariarita Brancaccio, Cristina Mennitti, Margherita G. De Biasi, Maria Elena Pero, Giuseppe Pisanelli, Olga Scudiero, Raffaela Pero

**Affiliations:** 1Department of Pharmacy, University of Naples Federico II, Via Montesano, 80138 Naples, Italy; slaneri@unina.it (S.L.); margherita.debiasi@unina.it (M.G.D.B.); 2Department of Molecular Medicine and Medical Biotechnology, University of Naples Federico II, Via S. Pansini 5, 80131 Naples, Italy; brancacciomariarita2@gmail.com (M.B.); cristinamennitti@libero.it (C.M.); 3Department of Veterinary Medicine and Animal Production, University of Naples Federico II, Via Federico Delpino 1, 80137 Naples, Italy; mepero@unina.it (M.E.P.); gpisanel@unina.it (G.P.); 4Ceinge Biotecnologie Avanzate S.C.aR.L., 80131 Naples, Italy; 5Task Force on Microbiome Studies, University of Naples Federico II, 80100 Naples, Italy

**Keywords:** antimicrobial peptide, defensin, COVID-19, physical activity, cathelicidin, chemokine receptor, toll-like receptor

## Abstract

Antimicrobial peptides (AMPs), α- and β-defensins, possess antiviral properties. These AMPs achieve viral inhibition through different mechanisms of action. For example, they can: (i) bind directly to virions; (ii) bind to and modulate host cell-surface receptors, disrupting intracellular signaling; (iii) function as chemokines to augment and alter adaptive immune responses. Given their antiviral properties and the fact that the development of an effective coronavirus disease 2019 (COVID-19) treatment is an urgent public health priority, they and their derivatives are being explored as potential therapies against COVID-19. These explorations using various strategies, range from their direct interaction with the virus to using them as vaccine adjuvants. However, AMPs do not work in isolation, specifically in their role as potent immune modulators, where they interact with toll-like receptors (TLRs) and chemokine receptors. Both of these receptors have been shown to play roles in COVID-19 pathogenesis. In addition, it is known that a healthy lifestyle accompanied by controlled physical activity can represent a natural weapon against COVID-19. In competitive athletes, an increase in serum defensins has been shown to function as self-protection from the attack of microorganisms, consequently a controlled physical activity could act as a support to any therapies in fighting COVID-19. Therefore, including information on all these players’ interactions would produce a complete picture of AMP-based therapies’ response.

## 1. Introduction

Antimicrobial peptides (AMPs) are crucial members of the innate immune defense against invading microorganisms, including viruses [1]. AMPs can modulate the innate defense functions of host cells and tissues due to their pleiotropic nature [2,3,4,5,6,7,8,9,10]. 

In humans, the most characterized classes of AMPs are the defensins (α- and β-defensins) and the cathelicidin LL-37 [11,12]. 

AMPs can directly kill microbes through various mechanisms, such as membrane permeation or the interruption of electrochemical gradients. Furthermore, these peptides can modulate host immune responses thanks to their ability to communicate with the innate and adaptive immune system: AMPs interact with innate and adaptive immune receptors, such as toll-like receptors (TLRs), chemokines, as well as those of inflammasomes and their complement systems [2,13]. These intricate interactions result in a link between innate and adaptive immunity. 

Defensins, crucial components of the innate immune system, play an important role against infections as part of the first-line immunity. In fact, their ability to fight viruses, including human immunodeficiency virus (HIV), herpes simplex virus (HSV), influenza and SARS-CoV has been widely demonstrated [14]. The relationship between SARS-CoV-2 infection and defensin is not yet clear. Some researchers have evaluated the expression of defensin genes in the mouth cavity during COVID-19 infection [15]. Results showed that defensin genes are expressed in the nasopharyngeal/oropharyngeal cavity and are significantly downregulated in SARS-CoV-2 infection patients compared to negative controls. Significantly downregulated genes were defensin beta 4A, 4B, 106B, 107B, 103A and defensin alpha 1B. Downregulation of defensin genes suggests that innate immunity may be impaired in SARS-CoV-2 infection, thereby promoting disease. Consequently, the upregulation of the defensin gene expression and use of defensin peptides could represent interesting therapeutic interventions [15,16].

In addition, it is well known that controlled physical activity and adequate nutrition can represent a natural defense of the body; in this case, both physical activity and nutrition can support the immune system, improving its efficiency and defending the body from external attacks [17,18,19,20,21]. In fact, previous studies [9] have shown that competitive athletes have high serum levels of α and β defensins guaranteeing protection from any pathogenic microorganisms [10].

In this *scenario*, this review is an overview of defensins by summarizing their potential antiviral role concerning coronaviruses, and exploring their treatment of patients with coronavirus disease 2019 (COVID-19). Furthermore, on one hand we briefly describe the importance of TLRs and chemokine receptors’ biological characteristics to understand AMP-based therapeutic outcomes better; on the other hand we will emphasize the key role of physical activity in understanding the self-defense mechanisms promoted by the human organism against the appearance of infections and the role played by AMPs in these biological processes.

## 2. Characteristics of Coronavirus

COVID-19 has come to humans via a spillover, a natural process in which a zoonic pathogen evolves and can infect, reproduce, and transmit within the human species. This evolutionary phenomenon occurs mainly in RNA viruses since they have a higher mutational rate than DNA viruses [22,23]. 

Coronaviruses possess a single-stranded linear RNA genome with positive polarity and a length of 27–32 Kb. They have a spherical shape with a diameter of about 200 nm, consisting of a phospholipid envelope, the pericapsid, with distal branches named peplomers, giving the virus a crown-like appearance (Figure 1). 

Coronaviruses are made up of several proteins: (i) N protein (N) stabilizes RNA; (ii) glycoprotein S (S) forms pleplomers and promotes virus attack and fusion with the host cell membrane; (iii) M protein (M) is an envelope protein necessary for virus assembly; (iv) protein E (E) is a component of the envelope; (v) hemagglutinin-HE (HE) is involved in the release of the virus (Figure 1). 

At the genotypic and serological level, CoVs are divided into four categories: α, β, ɣ, and δ. Additionally, β-CoVs are split into four other lines A, B, C, D. These viruses can provoke several respiratory conditions, ranging from the common cold to Middle East Respiratory Syndrome (MERS) to Severe Respiratory Syndrome (SARS) [24,25,26,27,28].

Significant common symptoms are fever, headache, cough, difficulty breathing, and diarrhea. In some cases, these symptoms may be silent; on the other hand, the manifestation is violent to the point of causing severe pneumonia, dyspnoea, renal failure, and even death [29,30,31,32].

During viral infections, the host activates the immune system to fight the pathogenic microorganism. An out-of-control immune response can occur during a violent infection such as COVID-19, resulting in substantial lung tissue damage [23,29,30].

### SARS-Cov-2 Infection

SARS-CoV-2 is an enveloped virus that possesses a characteristic spike glycoprotein (S) deputy for viral entry through the cell surface receptor of the angiotensin-2 converting enzyme (ACE2) [33]. Entry receptors’ expression and distribution in the host can influence the pathogenicity and tropism of the virus. The S1 subunit of protein S represents the receptor’s binding domain in ACE2 binding [34]. Subsequently, a cellular serine protease TMPRSS2 induces proteolytic cleavage of protein S at S1/S2 and S2 sites (protein S priming) [35]. 

TMPRSS2 is localized in the human respiratory tract and strongly contributes to both SARS-CoV-2 spread and pathogenesis. SARS-CoV-2 replicates abundantly in upper respiratory epithelia, where ACE2 is also expressed and is efficiently transmitted [33,34,35,36].

A particular characteristic of the SARS-CoV-2 S protein is acquiring a polybasic cleavage site (PRRAR) at the S1–S2 boundary, consenting to efficient cleavage by the pro-protein convertase furin. Cleavage results in enhanced infection and is a critical event in SARS-CoV-2 evolution and is the primary determinant in overcoming species barriers [34,35,36,37,38,39,40,41,42,43]. Furthermore, exposure of protein S on the virion surface results in the induction of peculiar neutralizing humoral immune responses [44]. Coronavirus S proteins are highly glycosylated and stimulate immune evasion by shielding epitopes from neutralizing antibodies [45,46,47]. However, sera from COVID-19 patients can neutralize SARS-CoV [33].

Many specific or cross-reactive antibodies can bind to the SARS-CoV S protein, and their administration to infected patients appears to provide immediate protection [43,48,49]. Furthermore, human monoclonal antibodies from preceding sets of hybridomas of transgenic mice, immunized with the SARS-CoV S protein, or the memory B cell repository of recovering patients with SARS and COVID-19 directly interpose with the RBD-ACE2 interaction or impair pre-fusion intermediate conformations on various binding epitopes [43,48,50,51].

After cleavage, both regions of the repeat 1 heptapeptide (HR1) and the repeat 2 heptapeptide (HR2) of the S2 subunit interact to form the fusion core of the six stranded bundle (6HB) [52,53]. The formation of 6HB promotes the viral membrane fusion procedure for viral ingress within the host cell by endocytosis. Along the late endosomal phase, endosomal acidification induces virus-endosomal membrane fusion, thus driving to a viral coating (release of viral RNA) to allow the starting of viral replication and infection [54,55,56]. 

During the intracellular life cycle, SARS-CoV-2 expresses and replicates its genomic RNA to generate full-length copies that are enclosed into only just-built viral particles. SARS-CoV-2 possesses large genomic RNA flanked by untranslated 5′ and 3′ regions containing secondary cis-acting RNA structures necessary for RNA synthesis.

At the 5′ end, the genomic RNA contains two sizeable open read frames (ORF; ORF1a and ORF1b). ORF1a and ORF1b encode 15–16 non-structural proteins (nsp), of which 15 constitute the viral replication and transcription complex (RTC) (RNA processing and modification enzymes and an RNA correction function necessary to maintain the integrality of the Coronavirus genome [57].

ORFs encoding structural proteins and interspersed ORFs encoding accessory proteins are transcribed from 3′ one-third of the genome to form a nested ensemble of subgenomic mRNA (sg mRNA). The ORFs that encode the structural proteins are (i) protein S, (ii) envelope protein, (iii) membrane protein, and (iv) nucleocapsid protein N, and are found in the 3′ one-third of the genomes of the coronavirus. Interviewed among these ORFs are the ORFs that code for accessory proteins. SARS-CoV-2 structural proteins have not yet been studied with regards to virus assembly and budding. Recent evidence shows that infected cells exit SARS-CoV-2 via the lysosomal trafficking pathway. Over this procedure, viral interference occurs with lysosomal acidification, lysosomal enzyme activity, and antigen presentation [58].

SARS-CoV-2 contains at least five ORFs that code for accessory genes: ORF3a, ORF6, ORF7a, ORF7b, and ORF8, as well as potentially ORF3b99 and ORF9b100 [52,53,54].

Furthermore, an ORF10 appears to be situated downstream of the N gene. However, not all ORFs have been scientifically studied, and the correct number of SARS-CoV-2 accessory genes is still debated [42,59,60].

The coronavirus’s accessory proteins are poorly preserved even within single species, but it seems that they mainly concur to regulating host responses to infection and are crucial for viral pathogenicity [61,62,63,64]. However, many accessory proteins’ molecular functions are widely hidden due to the absence of homologies with accessory proteins of other coronaviruses or with other noted proteins [65,66,67,68]. They are not utilized for virus replication in vitro, but, being preserved among their specific viral species, they are presumed of having essential roles in the natural host [62,63].

In addition, as of late, SARS-CoV-2 ORF8 has been described to link to a major histocompatibility complex and mediate its degradation in vitro [66,69,70,71].

## 3. Defensins: Alpha and Beta-Defensins

Defensins are a family of small cysteine-rich cationic and amphipathic peptides that belong to either the α or β subfamily. **Alpha-defensins** are abundant in neutrophils and Paneth cells of the small intestines. In neutrophils, these α-defensins are referred to as human neutrophil peptides (HNP)1, 2, 3, and 4. Enteric α-defensins are referred to as human defensins (HD)5 and HD6 [1,2,3]. Alpha-efensins have been isolated from neutrophils and, more specifically, within the azurophilic granules in which HPN1-3 represent approximately 50% of protein content, while HNP4 is present at low concentrations [72]. Defensins HNP1- and 3 are also present in B and natural killer lymphocytes.

Defensins HD5- and 6 are defined as enteric defensins because they have been found in Paneth cells of the small intestine and in the epithelial cells of the female urogenital tract (endometrium, and fallopian tubes) [73]. Alpha-defensins play a key role in oxygen-dependent destruction of phagocytised microorganisms. They are synthesized as pre-pro-peptides (93–100 aminoacids) with a 19-amino acid signal peptide and a 41–51 amino acid anionic pro-segment. The core of alpha-defensin molecules consists of three beta strands that connect cysteines 1–6, 2–4 and 3–5. In addition, in alpha-defensin, but not in the beta class, the beta sheet is flanked by an alpha helical segment of variable length, corresponding to the N-terminal domain. This particular conformation and charges distribution are probably responsible for the antimicrobial activity of defensins [74].

**Human β-defensins** (hBD)1, 2, 3, and 4 are expressed in various mucosal epithelial cells and participate in the mucosal innate immune defense against microbial colonization. Some β-defensins have also been identified in non-epithelial cells [75]. Non-epithelial expressed β-defensins can be crucial in the protection against viruses; for example, they have been identified in the antiviral plasmacytoid dendritic cells and other innate immune cells [76]. Beta-defensin proteins have been shown to reach biologically relevant concentrations in the serum and plasma [77]. The antimicrobial activity of defensins was initially attributed to the selective perturbation of microbial membrane lipids. For enveloped viruses, this membrane perturbation inhibits receptor binding and fusion of the virus to host cells [1]. However, there are several classes of non-enveloped viruses that are also sensitive to defensin actions [1]. Among those different mechanisms, downregulation of the viral receptor and disruption of the early molecular events in viral infection are included; however, a universal mechanism related to the neutralization of non-enveloped viruses is still unclear [78]. Defensins can interfere with intracellular steps of viral replication, which remain lesser-known [79].

Antiviral activities of defensins have been shown for various viruses such as herpesviruses, vaccinia virus, adenoviruses, papillomaviruses, polyomaviruses, influenza A virus, respiratory syncytial virus, human immunodeficiency virus, rhinovirus, and Aichi virus A. The antiviral activities for these viruses may differ between types (α or β) and subtypes (HNP1-4, HD5, HD6, hBD1-4) of defensins and between defensins [1,78,79].

## 4. LL37: Human Cathelicidin

Cathelicidin in humans is called human cationic antimicrobial protein (hCAP). The molecular weight of the inactive precursor protein is 18 kDa (hCAP18). Following processing, an active peptide of 37 amino acids is released, starting with a double leucine (LL-37). Neutrophils and epithelial cells mainly express LL-37 during acute inflammation [80,81]. After neutrophil activation, hCAP18 is enzymatically cleaved by a proteinase into the active form LL-37, released during degranulation processes [82]. Inflammation can greatly increase the concentration of LL-37 at the sites of inflammation. LL-37 can specifically permeabilize prokaryotic membranes and is directly microbicidal against bacteria, fungi, and many viruses, both with and without an envelope [80,81,82]. 

Although the mechanism of action for the direct microbicidal activity of LL-37 has not been definitively determined, the antimicrobial mechanism is likely to depend on the peptide’s particular membrane penetrating properties [80,81,82,83]. LL-37 can inhibit viral infection by targeting the steps that precede the virus entering the cell. It can: (i) create pores within the viral envelope; (ii) cause extracellular aggregation of viral particles that block the virus’s entry and increase the absorption of the virus by phagocytes; (iii) inhibit the attachment of the virus to its receptor on the cell surface [82]. In addition, LL-37 and defensins can interfere with viral replication’s intracellular stages [82]. A preliminary evaluation of the safety and efficacy of oral LL-37 (cas001), produced by genetically modified *Lactococcus lactis*, was performed in 11 COVID-19 patients with mild symptoms [83]. These patients were treated with cas001 for 3 weeks. All treated patients improved systemically, respiratory and gastrointestinal, and none experienced adverse reactions during this clinical study. While these results are promising, another evaluation in large-scale clinical trials is needed to assess the potential efficacy of cas001 [83]. 

Interestingly, an “in silico” study showed a high structural similarity of LL-37 to the N-terminal helix of the SARS-CoV-2 RBD [84]. Through molecular docking analysis, this study predicted that LL-37 would bind to RBD, proposing that LL-37 has the potential to block RBD: ACE2 interactions. However, the study also recognized that further experimental studies would be needed to evaluate the actual binding of LL-37 to the virus’s RBD [84] (Table 1).

Other evidence that LL-37 may be involved in protection against SARS-CoV-2 arises from its regulation by vitamin D. The active metabolite 1,25-dihydroxy stimulates transcription of the cathelicidin gene, and a positive association exists between circulating bioactive vitamin D and hCAP18/LL-37 levels in healthy subjects [93,94]. Vitamin D and LL-37 play an essential role in lung immunity and respiratory disease response [95]. Evidence suggests that blood levels and vitamin D bodies may be a risk factor for SARS-CoV-2 or COVID-19 disease [96,97,98,99]. Therefore, vitamin D deficiency can lead to reduced levels of LL-37 and a weakened antimicrobial response to SARS-CoV-2 [100]. Other studies found a correlation between lower vitamin D levels and mechanistic COVID-19 [101]. Hypothesis-testing studies are now needed to understand whether treatment of vitamin D deficiency would increase LL-37 levels and contribute to disease prevention to improve the prognosis of patients with COVID-19.

## 5. Defensins against Coronaviruses

Given their antiviral effects, AMPs represent promising therapeutic tools against emerging infectious viral pathogens, which have posed enormous challenges for vaccine development [11,102]. The use of human defensins as antiviral agents has been demonstrated in several animal models. These studies clarify the efficacy and the different mechanisms of different types of defensins against viral pathogens [102]. Furthermore, human defensins have a potential adjuvant activity of the vaccine [102]. Below, we summarize the studies demonstrating that defensins and their derivatives can be considered potential anti-severe agents of acute respiratory syndrome coronavirus 2 (SARS-CoV-2) through their different mechanisms of action. 

## 6. Mechanisms of Action of Defensins

Defensins can have direct and indirect actions. Direct antiviral and immunomodulatory potential of human defensins has been studied against coronaviruses as Middle East respiratory syndrome coronavirus (MERS-CoV) and severe acute respiratory syndrome coronavirus (SARS-CoV) [103].

### 6.1. Mouse β-Defensins-4 Derived P9

P9 derived from murine β-defensin-4 (an ortholog of hBD2) can bind to the MERS-CoV S2 subunit and remains co-localized with the virus in the host cell [104]. Within the endosomes, the polycationic properties of P9 create a basic microenvironment that inhibits the acidification of late endosomes. In the absence of endosomal acidification, pH-dependent activation of viral fusion proteins is prevented to initiate fusion of the viral host’s endosomal membrane. Consequently, the crucial step of removing the viral coat before the release of viral RNA is also impeded (Figure 2). 

This mechanism of action was highly influential, extensively inhibiting SARS-CoV, MERS-CoV, and a heterogeneous group of H1N1, H3N2, H5N1, H7N7, and H7N9 influenza viruses. This peculiar nature of the P9 target also in part elucidates the low cytotoxic feature of P9 versus mammalian canine kidney cells tested. The great therapeutic windows of P9 would depict a meaningful benefit in the coming evolution of P9 as an antiviral agent [104]. Additionally, four other defensin peptides [two rat α-defensin, one human α-defensin (HD5), and one chicken avian β-defensin] can also strongly bind to MERS-CoV S2 [105,106]. 

Prophylactic and therapeutic effects of P9 were examined in a BALB/c mouse model of non-lethal SARS-CoV infection by administering one dose and five doses of the peptide, respectively. This model observed significant viral infection inhibition in the treated mouse lungs at day 3 post-infection compared to that in the mouse’s untreated lungs [107].

Recently, it has been demonstrated that another short peptide, P9R, which has a more net positive charge compared with the parent peptide P9, not only had more potent antiviral activity against SARS-CoV and MERS-CoV but also SARS-CoV-2 [107].

### 6.2. HD5: Human Intestinal α-Defensin-5

SARS-CoV-2 protein S occupies a crucial position in the process of receptor recognition and cell membrane fusion [89]. The S1 subunit includes a receptor-binding domain (RBD) that identifies and engages the host receptor, the angiotensin 2 converting enzyme (ACE2) [90]. Researchers are exploring the possibility of preventing virus-induced death. SARS-CoV-2 S/ACE2 interactions for therapeutic purposes [106,107].

SARS-CoV-2 S1/ACE2 protein-protein interactions, necessary for viral invasion into host cells, is inhibited by human intestinal α-defensin HD5, a natural lectin-like human defensins-5 peptide secreted by the Paneth cells [108]. 

HD5 bound to ACE2 and competitively cloaked several sites in the ligand-binding domain that are crucial for SARS-CoV S/ACE2 interactions. Thus, SARS-CoV-2 S1 subunit binding and spike protein pseudovirions entry to enterocytes were inhibited by HD5 (Figure 1) [90]. The high-affinity binding among HD5 and the ligand-binding domain of ACE2 across the generation of manifold hydrogen bonds preserved the host cells from viral recognition and infection [108]. 

Interestingly, Junwen Luan et al. [108] found that high levels of DEFA5 were present in ACE2-expressing cells (small intestine enterocyte progenitor cell and small intestine enterocyte). Thus, DEFA5 in the human intestine could reduce the entry of SARS-CoV-2, which partially explains the low incidence of diarrhea in COVID-19 patients.

### 6.3. HBD2: Human Beta Defensin 2

HBD-2 is expressed all around the respiratory epithelium from the oral cavity to the lungs and plays a crucial role against respiratory infections [91]. Altered hBD-2 expression in the respiratory epithelium is related to the pathogenesis of different respiratory diseases, such as asthma, pulmonary fibrosis, pneumonia, tuberculosis, and rhinitis [91,92,109,110]. HBD-2 can repress the human respiratory syncytial virus (RSV) infection by stopping viral entry across destabilization/disintegration of the viral envelope [111].

It might have critical immunomodulatory functions in the course of coronavirus infection; in fact, hBD-2 conjugated to the MERS receptor-binding domain (RBD) has been demonstrated in a mouse model to encourage finer protective antibodies to RBD than RBD alone. Zhang et al. [89] studied the capacity of hBD-2 to act as an inhibitor factor against CoV-2. HBD-2 is an amphipathic, beta-sheeted, extremely cationic molecule of 41 amino acids and is stabilized by three intramolecular disulfide bonds that preserve it from degradation by proteases [112,113]. The protein has been examined before through molecular dynamics simulations [114,115]. Zhang et al. [89] observed that HBD-2 could bind the receptor-binding motif (RBM) of the RBD of CoV-2 that links to the ACE2 receptor. Biochemical and biophysical analysis validated that HBD-2 binds the RBD and also avoids it from binding ACE2. 

Moreover, using a physiologically relevant shelf, they demonstrated that HBD-2 inhibits CoV-2 spike from invading ACE2 expressing human cells. Taking advantage of the utility of natural AMPs, such as hBD-2 and their derived smaller peptides, could be a viable strategy to generating novel CoV-2 therapeutics.

### 6.4. RTD-1: Rhesus θ-Defensin 1

The rhesus θ-defensin 1 (RTD-1), a cyclic AMP, was an extremely valid preventive antiviral in a mouse model of severe SARS-CoV lung disease [116]. BALB/c mice infected with a mouse-adapted strain of SARS-CoV and dealt with two intranasal doses of RTD-1 manifested 100% survival, while mortality in untreated mice was ~75% [116]. RTD-1 was proposed to work as an immunomodulatory effector molecule by a straightforward proinflammatory cytokine response in removing SARS-CoV. These findings suggest that one potential mechanism of action for RTD-1 was immunomodulatory. Theta-defensin peptides are produced only in old-world monkeys and orangutans. In humans, they are expressed as pseudogenes owing to the presence of an early stop codon [117]. Synthetic θ-defensin peptides, whose sequences coincide with those encoded within the human pseudogenes, are called retro-cyclins. They are active against HIV-1, influenza A virus, and herpes simplex viruses [118]. The use of aminoglycoside antibiotics can bring termination codon readthrough of retrocyclin cDNA [111]. In further experiments, using human cervico-vaginal tissue models, it was revealed that aminoglycosides could re-establish the translation and anti-HIV-1 activity of native retrocyclin peptides [118]. Given that θ-defensin peptides are active against SARS-CoV lung disease, as well as against HIV-1, influenza A virus, herpes simplex viruses, and consideration of aminoglycosides induce the production of these AMPs against SARS-CoV-2 could be beneficial [117,118].

## 7. Interplay between Physical Activity, Human Defensins and Immune System

Moderate and controlled physical activity is the body’s natural weapon against cellular aging, in fact physical activity is recommended at any age.

Furthermore, when physical activity is supported by proper nutrition, it can be a shield against the onset of infections and diseases [19,20].

Practicing sports regularly and constantly lowers the risk of contracting respiratory diseases and generally increases the body’s immune defences [21].

Recent studies have shown that competitive athletes between 3 and 6 months of activity during the championship have an increase in serum of α and β defensins and that this increase was useful for athletes as self-defense, in fact, they showed biochemical parameters such as PCR and normal white blood cells [9].

At the same time, it has been described in literature that defensins can protect athletes from the appearance of respiratory and skin infections caused by microorganisms such as *S. aureus* [10].

Consequently, moderate physical activity would help the immune system to defend itself against COVID-19 by increasing the expression of certain innate immune protagonists such as defensins.

## 8. Vitamin D Generates Defensins Active against COVID-19

Vitamin D determines the expression of some antimicrobial peptides (defensin beta 2 and cathelicidin), which can act against enveloped viruses, including SARS-CoV-2. These could be the mechanisms that clarify the possible preventive result of vitamin D against COVID-19 [109]. 

Advancing gene transcription of antimicrobial peptides required binding of calcitriol and its receptor to a gene promoter region, which for vitamin D is VDRE [96,97].

Furthermore, the activation of toll-like receptors (TLRs) on the cell surface of human macrophages by the tuberculosis protein induced the upregulation of genes driving the yield of the vitamin D receptor and the enzyme vitamin D-1-hydroxylase. [97]. This enzyme transforms circulating 25 (OH) D into calcitriol on request within cells and augments the vitamin D-regulated genes that produce the antimicrobial peptide cathelicidin, killing intracellular Mycobacterium tuberculosis.

These data suggest a correlation between TLRs and vitamin D-mediated innate immunity and that difference in the incapacity of humans to make vitamin D may concur to predisposition of microbial infections [97]. The same is now happing with the coronavirus pandemic. Non-Hispanic blacks are notably more lacking in vitamin D [119,120] and suffer more severe complications and deaths from coronavirus infections than other groups [120,121]. This relation squares with many late studies, which have demonstrated that vitamin D has a critical role in modulating the immune response against viral infections [122,123,124,125,126,127,128,129,130,131] and that deficiency allows elevated susceptibility to respiratory infections [132,133]. 

Moreover, short alterations in VDR alleles that influence activity can affect the susceptibility or resistance to infection. A report of Canadian children has demonstrated that VDR gene polymorphisms increment the probability of severe lower respiratory tract infections [131].

Vitamin D deficiencies can affect viral pathogenesis. Once a cell is prompted by microorganisms linking with toll-like receptors on its surface, elevated production of the VDR and the enzyme 25-hydroxyvitamin D 1-alpha hydroxylase (which converts circulating 25(OH)D into calcitriol intracellularly) are revealed [93]. Subsequently, calcitriol is produced, on the requirement within cells, and links to the VDR and several other transcription factors. During this process the whole complex is linked to the VDRE, allowing the up or down-regulation of genes modulated by vitamin D. Some of these genes render antimicrobial peptides capable to kill manifold microorganisms [134,135,136,137]. 

Thus, the supplementation with high doses of vitamin D that effectively treat tuberculosis infections could be due to increased AMP production [138,139].

## 9. Other Therapeutic Defensins

Recently, research exploring the therapeutic application of mesenchymal stem cells (MSCs) in critical COVID-19 patients has gained momentum [140,141]. Several studies have indicated that an uncontrolled host immune response, leading to a life-threatening condition called “cytokine storm,” is the primary driver of pathology in severe COVID-19 [142]. Because MSCs possess inherent regeneration and repair properties, current efforts focus on their capability to abort or minimize this “cytokine storm,” thereby reducing lung damage and promoting tissue function restoration [142,143]. In addition to their immunomodulatory properties, MSCs possess antimicrobial properties, mediated partly by the secretion of LL-37, hBD2, and other AMPs [143,144,145]. Most of the data about the antimicrobial properties of MSCs have been obtained from studies with bacteria, and little data exist about the effect of MSCs on viruses and other pathogens [143]. Considering that certain AMPs have antiviral and, more importantly, the potential for anti-coronavirus activity, we surmise that AMPs may partly mediate the effect of MSCs on SARS-CoV-2. However, other studies are needed to prove this conjecture.

Brilacidin (PMX-30063), a fully synthetic non-peptide mimetic of defensins, is being tested as a potential treatment for COVID-19 [146]. The investigational drug has shown efficacy against SARS-CoV-2 in Vero cells (40% and 50% reduction at 2 and 10 μM, respectively). As Brilacidin exhibits antiviral, immunomodulatory/anti-inflammatory, and antimicrobial therapeutic properties and has been tested successfully in Phase 2 clinical trials for other infectious/inflammatory diseases, it is considered a promising and novel coronavirus therapeutic candidate [146].

## 10. Defensins as Vaccine Adjuvants

AMPs also have the potential to be explored as vaccine adjuvants against coronaviruses [146,147,148,149]. C57BL/6 mice immunized with 10 μg of the hBD2-conjugated MERS-CoV S RBD had significantly greater S RBD-specific IgG levels compared with those receiving S RBD alone [147].

A multi-epitope subunit vaccine has been constructed to elicit the immune response against SARS-CoV-2, which was comprised of HBD3 as an adjuvant at the N-terminal end followed by B cell, helper T lymphocyte, and cytotoxic T lymphocyte epitopes [147]. A stable and robust interaction between the vaccine candidate and binding groove of the immune cell receptor TLR3 was observed by molecular docking and molecular dynamics simulation [149]. 

Immune stimulation, mimicking the natural immune environment, showed proliferation of the immune response, an active B-cell population, generation of helper and cytotoxic T lymphocytes, and induced and retained IFN-γ levels in response to the vaccine candidate [149]. These in silico experiments provide encouraging data to pursue further immunological studies.

## 11. Conclusions

The immune response to SARS-CoV-2 infection is being characterized, and considerable effort is being focused on identifying host factors that increase susceptibility or resistance to the complications of COVID-19. To improve patient care, it seems reasonable to explore the role of AMPs in COVID-19 pathogenesis and treatment. 

Given the massive effort that the biomedical research community is investing into finding drugs to treat COVID-19 [144], with a vast number of trials completed and ongoing, it is worth examining the evidence for AMP-based therapies. So far, the evidence is promising. 

AMPs can act both directly and indirectly against coronaviruses [23,24,27]; they can act as protein-protein interaction inhibitors to block cell invasion by the virus [127,128]; LL-37, either alone, due to regulation by vitamin D, or due to an antimicrobial property of MSCs, which is emerging as another therapeutic option [141,142,143].

Fully synthetic non-peptide mimetics of defensins are promising drug candidates; and finally, they can be used as adjuvants in vaccines targeting coronaviruses [137,148,149,150,151].

AMPs do not work in isolation, specifically in their role as potent immune modulators. These peptides can induce chemokine release and enhance the recruitment of leukocytes, and this ability defines a primary immunomodulatory mechanism by which they protect against infections [1,2,3]. The underlying immunomodulatory agent involves their interactions with various cellular receptors, including TLRs and chemokine receptors [2,3,10]. 

AMPs can modulate TLR expression and functionality [93,94]. Inversely, TLRs have been shown to mediate the expression of hBDs in various tissues [121,122]. Also, the ability of hBDs to cross-talk with chemokine receptors has been reported [122]. Although much is yet to be understood, based on current knowledge, TLRs and the chemokine receptor-ligand nexus seem to be involved in COVID-19 disease pathogenesis [127]. In an “in silico” study, cell surface TLRs, mainly TLR4, were most likely engaged in recognizing molecular patterns from SARS-CoV-2 that induce inflammatory responses [129]. 

Finally, AMPs can be stimulated by factors such as physical activity, which promote its expression. This phenomenon would be interesting in evaluating it also in subjects in post-COVID rehabilitation, to verify whether individuals who present an early recovery are individuals who conduct styles of healthy living, and therefore are also benefited by the presence of elevated serum defensin levels when compared with subjects who are not physically active.

Many strategies are being used to achieve the practical and successful application of AMPs, their derivatives, and AMP-based therapies against COVID-19. These strategies, alone or in combination with one another, may complement other COVID-19 treatment-related efforts. The urge to find effective treatments is now more pressing than ever since the SARS-CoV-2 outbreak became a pandemic.

In conclusion, AMPs are certainly valuable molecules, thanks to their multiple properties. Through our overview we wanted to underline how these molecules can be a further weapon against COVID-19.

## Figures and Tables

**Figure 1 microorganisms-09-01415-f001:**
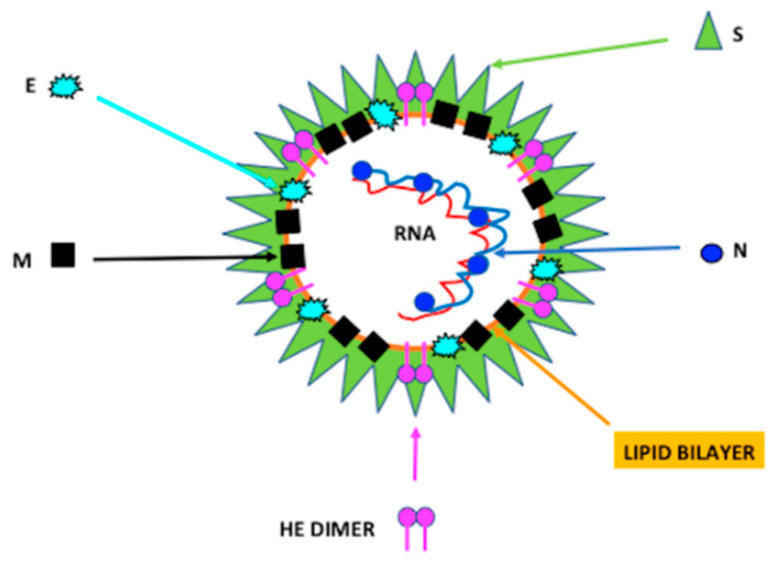
Characteristics of coronavirus.

**Figure 2 microorganisms-09-01415-f002:**
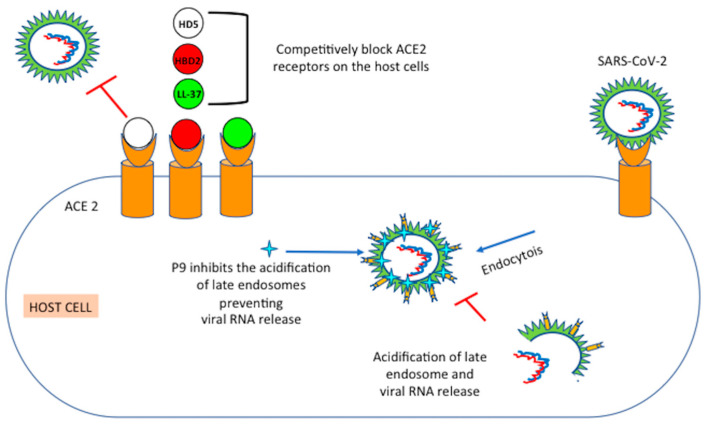
Mechanism of actions of some defensins with potential anti-SARS-CoV-2 activities. P9 peptide inhibits late endosomal acidification preventing viral RNA release. HD5, LL37 and HBD2 bind to ACE2 blocking its viral recognition and binding.

**Table 1 microorganisms-09-01415-t001:** Characteristics of defensins and LL-37 with potential inhibitory activity against Sars-Cov2.

Peptide	Sequence	Source	Inhibitory Concentration	Cell Line	Ref.
LL-37	LGDFFRKSKEKIGKEFKRIVQRIKDFLRNLVPRTE	Neutrophils and epithelial cells	100 ng/mL	Vero E6 cells	[82,83,84]
P9	NGAICWGPCPTAFRQIGNCGHFKVRCCKIR	Mouse β-defensin-4	~5 μg/mL (IC50) 25 μg/mL (IC90)	FRhK-4 and Vero-E6 cells/BALB/c mice	[85]
P9R	NGAICWGPCPTAFRQIGNCGRFRVRCCRIR	Mouse β-defensin-4	0.9, 2.2 and 4.2 μg/mL (IC50)	Vero E6 cells	[86]
HD5	ATCYCRTGRCATRESLSGVCEISGRLYRLCCR	Intestinal Paneth cells and neutrophils	10 μg/mL	Caco-2 cells	[87]
HBD2	GIGDPVTCLKSGAICHPVFCPRRYKQIGTCGLPGTKCCKKP	Surface of epithelial barriers	IC 50 of 2.4+/−0.1 μM	ACE2 HEK 293T cells	[88]
RTD1	GFCRCLCRRGVCRCICTR	Rhesus macaque leukocytes	--------	-------	[89,90,91,92]

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
