# Peer review of "Antimicrobial Peptides and Physical Activity: A Great Hope against COVID 19"

_microorganisms, 2021, doi:10.3390/microorganisms9071415_

Round 1

Reviewer 1 Report

The authors have tried to review the possible role of α- and β-defensins and its usefulness to treat the COVID 19.
This article can be accepted after a few changes.

1. A proper graphical abstract is needed for this review to denote the exact mechanism. 
2. There are lots of studies now available on the same topic . Authors need to include the scientific impact of this study in the Introduction.
I suggest to include the following papers concepts https://doi.org/10.1016/j.jaci.2020.05.033
3.  In discussion, mention what knowledge gained from this review  to the readers
4. Alpha-defensins promote tumour cell growth or, at higher concentration, provoke cell death. Defensins inhibitors are used to treat different cancer patients. In the discussion section authors their strategies and COVID 19 infected cancer patients. Refere the article which discuss about the topics http://waocp.com/journal/index.php/apjcc/article/view/508 5. To increase the value and audience for this article, the author should include a graphical abstract which can give an overall idea of this study.   

Author Response

REV1

Comments and Suggestions for Authors

The authors have tried to review the possible role of α- and β-defensins and its usefulness to treat the COVID 19.
This article can be accepted after a few changes.

1. A proper graphical abstract is needed for this review to denote the exact mechanism. 

  1. There are lots of studies now available on the same topic . Authors need to include the scientific impact of this study in the Introduction.
    I suggest to include the following papers concepts https://doi.org/10.1016/j.jaci.2020.05.033
    3.  In discussion, mention what knowledge gained from this review  to the readers
    4. Alpha-defensins promote tumour cell growth or, at higher concentration, provoke cell death. Defensins inhibitors are used to treat different cancer patients. In the discussion section authors their strategies and COVID 19 infected cancer patients. Refere the article which discuss about the topics http://waocp.com/journal/index.php/apjcc/article/view/508 5. To increase the value and audience for this article, the author should include a graphical abstract which can give an overall idea of this study.   

Thanks to reviewer 1 for the comments.

To satisfy your request, we have added a graphical abstract; also we added the reference https://doi.org/10.1016/j.jaci.2020.05.033 in the section 2; finally, we improved the introduction and conclusion.

Reviewer 2 Report

In this manuscript, authors described various aspects of antimicrobial peptides (not limited to defensins) relevant to SARS-CoV-2 infection. Although the topic is interesting, the manuscript seems to be at the preliminary stage. It is not well organized. There is too much incorrect information, in part because authors did not follow the original articles. The references are either missing or incorrectly cited. SARS-CoV and SARS-CoV-2 have distinct pathogenesis. It’s not appropriate to state the knowledge from SARS-CoV as SARS-CoV-2. The followings are specific comments.

  1. The title needs to be changed as the article is focused on antimicrobial peptides rather than defensins

  1. In Fig 1, the scheme for the M protein is incorrect. The M protein is an envelope protein, which should be drawn in a similar fashion as the E protein. It does not interact with nucleocapsid as described in line 67, page 2. Authors confuse the membrane protein in coronaviruses with matrix proteins in other RNA viruses.

  1. In page 3-4, line 109-124, the description is for SARS-CoV not SARS-CoV-2.

  1. In line 126-128, the abbreviation for the viral proteins should have defined in line 65 when first mentioned.

  1. The section 3.1 for alpha-defensin is poorly written.

  1. The section for beta defensin is not limited to beta defensins.

  1. It is incorrect regarding the anti-viral mechanism against non-enveloped viruses based on the work done by Jason Smith. Authors should read reviews by J. Smith or T. L. Chang.

  1. The reference 73 does not support statements in Line 193-197. Authors should read original articles rather than copying from other reviews.

  1. In line 206, Ref 75 has nothing to do with the interaction between LL37 and SARS-CoV-2 RBD.

  1. In line 287, ref 92 was cited in correctly. RTD does not inhibit SARS-CoV virus but prevents virus-induced death.

  1. There is no reference for the statements in line 347-351. There are many beta defensins. Authors should be more specific regarding the beta defensins regulated by VitD.

  1. Conclusion section is too long. Some materials should be in the other sections.

Author Response

REV2

Comments and Suggestions for Authors

In this manuscript, authors described various aspects of antimicrobial peptides (not limited to defensins) relevant to SARS-CoV-2 infection. Although the topic is interesting, the manuscript seems to be at the preliminary stage. It is not well organized. There is too much incorrect information, in part because authors did not follow the original articles. The references are either missing or incorrectly cited. SARS-CoV and SARS-CoV-2 have distinct pathogenesis. It’s not appropriate to state the knowledge from SARS-CoV as SARS-CoV-2.

The followings are specific comments.

We thank reviewer 2 for the helpful suggestions.

  • The title needs to be changed as the article is focused on antimicrobial peptides rather than defensins
  • We have now changed the title to “Antimicrobial peptides and physical activity: a great hope against COVID 19.
  • In Fig 1, the scheme for the M protein is incorrect. The M protein is an envelope protein, which should be drawn similarly as the E protein. It does not interact with nucleocapsid as described in line 67, page 2. Authors confuse the membrane protein in coronaviruses with matrix proteins in other RNA viruses.

We have now corrected the scheme for the M protein.

  • In page 3-4, line 109-124, the description is for SARS-CoV not SARS-CoV-2

We have now corrected SARS-CoV-2 in SARS-CoV

  • In line 126-128, the abbreviation for the viral proteins should have defined in line 65 when first mentioned.

We have now defined the abbreviation for the viral proteins in line when first mentioned.

  • The section 3.1 for alpha-defensin is poorly written.

We have now amplified the section 3.1

  • The section for beta defensin is not limited to beta defensins.

We have grouped both alpha and beta-defensins into a single section

  • It is incorrect regarding the anti-viral mechanism against non-enveloped viruses based on the work done by Jason Smith. Authors should read reviews by J. Smith or T. L. Chang.

We have read the cited reviews

  • The reference 73 does not support statements in Line 193-197. Authors should read original articles rather than copying from other reviews.

We have now changed reference 73 Chessa C, Bodet C, Jousselin C, Wehbe M, Leveque N, Garcia M. Antiviral and immunomodulatory properties of antimicrobial peptides produced by human keratinocytes. Front Microbiol. 2020;11:1155.

  • In line 206, Ref 75 has nothing to do with the interaction between LL37 and SARS-CoV-2 RBD.

We have now changed REF 75  Wang C, Wang S, Li D, Chen P, Han S, Zhao G, Chen Y, Zhao J, Xiong J, Qiu J, Wei DQ, Zhao J, Wang J. Human Cathelicidin Inhibits SARS-CoV-2 Infection: Killing Two Birds with One Stone. ACS Infect Dis. 2021 Apr 14:acsinfecdis.1c00096. doi: 10.1021/acsinfecdis.1c00096.

  • In line 287, ref 92 was cited in correctly. RTD does not inhibit SARS-CoV virus but prevents virus-induced death.

 We have now cited Ref 92 correctly

  • There is no reference for the statements in line 347-351. There are many beta defensins. Authors should be more specific regarding the beta defensins regulated by VitD.

We have now added reference for the statements in line 347-351 and specified that defensin beta 2 is regulated by Vit. D

  • Conclusion section is too long. Some materials should be in the other sections.

We have now reduced the Conclusion section

Reviewer 3 Report

I suggest the authors change the title to a more informative one. My main doubt regarding the work is the fact that the research presented by the authors is largely related to the role of defiensins in the pathophysiology of other infectious or pro-inflammatory diseases. Therefore, I believe that the authors should clearly define, preferably in the introduction to this paper, that the role of defensins in COVID-19 is rather speculative, and there is a need to conduct comprehensive in vitro and in vivo studies, but dedicated exclusively to SARS-CoV-2. This fact should reverberate very strongly.The further description of the role of the defensins is correct, detailed and does not raise any major reservations.
Figures - could be of higher quality.
References to be included in the manuscript:

doi: 10.3390/pathogens9060493;

doi: 10.3390/pathogens9070519;

 doi: 10.3390/pathogens9030231;

doi: 10.1055/s-0040-1718735. 

Overall, interesting paper worth publishing.

Author Response

REV3

I suggest the authors change the title to a more informative one.

Response:

Thanks to reviewer 3 for the comments.

To satisfy your request, we have changed the title.

My main doubt regarding the work is the fact that the research presented by the authors is largely related to the role of defiensins in the pathophysiology of other infectious or pro-inflammatory diseases.

Therefore, I believe that the authors should clearly define, preferably in the introduction to this paper, that the role of defensins in COVID-19 is rather speculative, and there is a need to conduct comprehensive in vitro and in vivo studies, but dedicated exclusively to SARS-CoV-2.

Response:

Thanks to reviewer 3 for the comments.

To satisfy your request, we have added a paragraph in the introduction section.

This fact should reverberate very strongly.

The further description of the role of the defensins is correct, detailed and does not raise any major reservations.

Figures - could be of higher quality.
References to be included in the manuscript:

  1. doi: 10.3390/pathogens9060493;
  2. doi: 10.3390/pathogens9070519;
  3. doi: 10.3390/pathogens9030231;
  4. doi: 10.1055/s-0040-1718735. 

Thanks to reviewer 3 for the comments.

To satisfy your request, we have added the suggested references.

Round 2

Reviewer 3 Report

The authors have addressed all the comments of the reviewer and revised the manuscript accordingly.